# Applications of Microalgae in Foods, Pharma and Feeds and Their Use as Fertilizers and Biostimulants: Legislation and Regulatory Aspects for Consideration

**DOI:** 10.3390/foods12203878

**Published:** 2023-10-23

**Authors:** Min Su, Leen Bastiaens, Joran Verspreet, Maria Hayes

**Affiliations:** 1The Food BioSciences Department Ashtown, Teagasc Food Research Centre, 15D05 Dublin, Ireland; min.su@teagasc.ie; 2Flemish Institute for Technological Research (VITO), 2400 Mol, Belgium

**Keywords:** application, bioactive compounds, legislation, microalgae, wastewaters

## Abstract

Microalgae are a rich resource of lipids, proteins, carbohydrates and pigments with nutritional and health benefits. They increasingly find use as ingredients in functional foods and feeds as well as in cosmetics and agricultural products including biostimulants. One of their distinct advantages is their ability to grow on wastewaters and other waste streams, and they are considered an environmentally friendly and cheap method to recover nutrients and remove pollutants from the environment. However, there are limits concerning their applications if grown on certain waste streams. Within, we collate an overview of existing algal applications and current market scenarios for microalgal products as foods and feeds along with relevant legislative requirements concerning their use in Europe and the United States. Microalgal compounds of interest and their extraction and processing methodologies are summarized, and the benefits and caveats of microalgae cultivated in various waste streams and their applications are discussed.

## 1. Introduction

It is predicted that demand for food will increase from 35% to 56% between 2010 and 2050 [1,2]. Consumers’ demand for meat and dairy products will increase by over 70%. However, at present, over 800 million people are malnourished [3]. Agriculture is necessary for food production but global annual greenhouse gas (GHG) emissions from traditional food systems are now known to account for around 34% of total global annual GHG emissions [4]. A sustainable future food supply in the face of the growing population, natural resource depletion, climate change and rapid urbanization is a global challenge and there is a need to add alternative biomasses to help feed our growing animal and human population and ensure food security.

Products derived from microalgae can contribute to food security and may help prevent environmental problems like greenhouse gas emissions (GHGs) associated with animal-sourced proteins. When in perfect environmental conditions, microalgae can grow extremely fast and it is estimated that the growth of algae can be up to 10 times quicker than land-based, conventional crops [5], making them a potentially sustainable source of feed ingredients. Microalgae use as food is not new and dates back to Aztec times but there is growing demand for food ingredients [6], feed ingredients [7], and agents for water purification [8] that are of algal origin. Moreover, they are known for their potential use as biofuels in renewable energy [9], as well as ingredients for use in biostimulants [10], cosmetic compounds [11] and healthcare products. For example, the alga *Parachlorella kessleri* R-3 is known to accumulate lipids and holds potential for use as a biodiesel when grown under suitable conditions [12]. Microalgae have advantages over terrestrial sources of ingredients due to their high growth rate (over 1 d^−1^) and high biomass productivity and yield [13] as well as their composition of bioactive compounds. Indeed, market demand for microalgae is set to be USD 55.67 billion dollars by the end of 2031 [14].

Microalgae have an excellent nutritional composition and content of proteins, lipids, and carbohydrates along with many therapeutically active enzymes, pigments, sterols, and vitamins [15,16,17]. Microalgal protein is reported to be similar to traditional protein sources like egg protein in terms of total amino acid composition [18]. A number of studies [15,19] have identified bioactive compounds from microalgae with benefits for human health including anticancer [20,21], anti-inflammatory [20,22], antimicrobial [23,24], antioxidant [22,25], and anti-obesity activities [26,27] as well as anti-hypocholesterolemia benefits [28]. Hence, they have potential for use as health-beneficial ingredients and nutraceuticals.

Overuse of antimicrobials including antibiotics for the treatment and prevention of animal illness has led to antibiotic resistance that affects both animal health and the food chain. For example, antibiotic use in feed as growth promoters has resulted in the development of antibiotic-resistant strains and antibiotic residues in food products destined for the human market previously [29]. Zinc oxide (ZnO), a regularly used antimicrobial, was recently banned for use in the EU due to the risk of absorption and subsequent accumulation in animals with potential to cause environmental contamination in the food chain [30]. Its use was examined by the EU in 2017 and drugs containing greater than 3000 mg Kg^−1^ of ZnO were banned for use in the EU to treat animals from 2022 [31]. Microalgal derived compounds have potential for use in animal as therapeutic agents due to their known bioactivities that include immunomodulatory [32], antioxidative [25], antimicrobial and antiviral benefits [33]. 

In terms of their use as foods, *Spirulina* sp. and *Chlorella* sp. in the form of dried, whole microalgal biomass are used as a food source currently in the EU and have applications for human health. In addition, bioactive extracts derived from other microalgae are used in different ways to provide a health benefit to consumers. For example, the immunomodulatory compounds sulphate polysaccharides derived from algae are promising candidates for drug development. In addition, sulfolipids are used as vaccine adjuvants to improve immune response against cancer cells [34]. The pigment astaxanthin is a known health food supplement, and commercially available astaxanthin derived from the microalga *Haematococcus pluvialis* has FDA approval for use [35]. The quantity of microalgal nutrients and their bioactive composition must be measured accurately if they are to be used as foods. Accurate methods for measurement of protein, lipid, vitamin and mineral contents are available, and these are nutritionally important components of microalgae that can benefit health [36]. The late-logarithmic growth phase of microalgae is the best time to harvest most algae as this phase results in more protein (30–40% more), about 10–20% lipids and 5–15% carbohydrates [37]. When cultured through to the stationary growth phase, composition changes. Changing the cultivation conditions for different species can also influence the nutrient and bioactive composition. Generally, lipid, protein and carbohydrate content can reach up to 77% dry weight [38], 70% dry weight [39] and about 50% dry weight, respectively. The Generally Recognized As Safe (GRAS) microalgal species including *Arthrospira platensis*, *Haematococcus pluvialis*, *Dunaliella bardawill*, *Chlorella protothecoides*, *Crypthecodinium cohnii* and *Porphyridium cruentum* contain about 40% protein. Soy contains around 38% protein and rice contains about 10% protein. Legumes like peas have up to 2.8% protein. Animal proteins like milk contain 4% protein while eggs are reported to contain 13% [40]. Algal protein is rich in essential amino acids compared to common plant proteins. Unlike most plant proteins, all GRAS microalgae with the exception of *Euglena gracilis* are complete protein sources [36]. In addition, microalgae contain bioactives like carotenoids, fatty acids, polyhydroxyalkonates and carbohydrates, and oligosaccharides, which can be extracted from algae and have potential for use in foods or functional foods as well as feeds [19,41,42].

Whilst microalgae promise to deliver many benefits, there are also several hurdles to overcome concerning their use as food and feed ingredients. Some of the major bottlenecks limiting the expansion of microalga use include costs and limitations in terms of the scale of production of microalgae. Microalgal production is still a small-scale activity because the high growth rate and productivity often observed at the laboratory scale is difficult to replicate at an industrial scale. Using outdoor cultivation conditions, the duplication time for microalga and the biomass productivity decreases (by as much as 40 t ha^−1^ per year) [43,44,45,46] compared to what is observed in the laboratory. Productivity of algae in terms of growth is usually lower in outdoor environments compared to laboratory environments because the temperature and light intensities outdoors are more variable and extreme [47]. Microalgae find it challenging to cope with natural fluctuations in unpredictable outdoor weather, especially for commercial production. Additionally, cultivation of algae outdoors in pond systems can result in environmental contamination with bacteria, virus, fungi, protozoa, rotifers or other unwanted algae if ponds are poorly designed and maintained. It is therefore important that the end biomass produced be assessed for the accumulation of toxins and pathogens especially where algae are produced on “waste” resources or wastewaters. The initial growth environment can therefore limit use of algae especially for feed, food and pharmaceutical purposes. Growth conditions also impact the potential to use algae as fertilizers and even their potential to be used as fuel feedstock as heavy-metal bioaccumulations could affect fuel properties and the composition of emissions [46].

Despite challenges, currently, global microalgal production is over 7000 tons (dry weight) per year. Microalgae produced today is used primarily for feed premixes and food applications [44]. The widely consumed microalgae worldwide include species belonging to the Genera *Spirulina*, *Dunaliella*, *Haematococcus* and *Chlorella*. Species belonging to these groups are used as foods, in tablets and supplements. Phycocyanin extracted from *Spirulina* sp. costs approximately 11 EUR mg^−1^ dry weight (DW) [43]. Beta-carotene extracted from *Dunaliella* sp. is priced at between 215 and 2150 EUR kg^−1^ DW [43]. Astaxanthin extracted from *Haematococcus* sp. can reach prices up to 7150 EUR kg^−1^ DW [43]. The fisheries sector generated greater than 7000 kilotons (kT) a year of fish oil and fish meal, used primarily for aquaculture feed [43]. More than 200,000 kT per year of soy oil and soy meal are produced and this resource contributes significantly to animal feed production globally. Prices for soy meal and soy oil regularly cost less than 0.5 EUR kg^−1^ [43]. This indicates that cost due to production scale and volume and additionally market uptake are hurdles towards widespread microalgal use. Additional issues concerning microalgal use include the problem that when operating at an industrial scale, cultivation of microalgae may be subject to contamination issues with biological pollutants. Growing microalgae is a symbiotic system where microalgae and bacteria as well as zooplankton communicate and assist each other’s growth. As a result, contamination often occurs and pure microalga culture production is costly [45]. Cultivation in open pond systems can result in contamination with environmental pollutants and culture collapse, resulting in loss of the product. In addition, there is the possibility of biomass contamination with pathogens present in harvested biomass or in the final process effluent when microalgae are cultivated with wastewater sources. This may present a potential health risk [46]. In addition, microalgae producers must become knowledgeable on the legislation governing the use of microalgae as/or in food and feeds, and they must be compliant with what is required to obtain an ecological footprint certification, and the Nagoya Protocol.

## 2. Current Markets for Microalgae as Food

### 2.1. Current Market Scenario

Valuable components can be extracted from microalgae and find applications in nutraceuticals or functional foods [48]. These extracts and bioactives are used in the formulation of soups, juices, biscuits, ice-creams and as natural coloring agents (Table 1), [49,50].

Nutrients and bioactive compounds including proteins and hydrolysates, as well as fatty acids, oligosaccharides and other small molecules contribute to health when consumed due to antioxidant, anti-inflammatory and other bioactivities [50]. 

Microalgae used in food and feed applications must first and foremost be safe for consumption and must be free of contaminants (like heavy metals), hazardous substances and must not pose a risk of causing allergy. Microalgae currently used for food and feeds are listed in Table 1 and are considered safe for use. 

At the moment, there are five microalgae-derived components—astaxanthin, β-carotene, phycocyanin, Omega-3, and two algae biomass products, i.e., *Spirulina* and *Chlorella* sp. approved for use as food and feed ingredients even though research has demonstrated the potential of a myriad of other microalgae for use in the food and feed industries. Table 2 lists different algal components with potential use as techno-functional or bioactive ingredients to improve nutrition and the health benefits of foods. 

### 2.2. Microalgal Compounds of Interests and Current Applications

#### 2.2.1. Pigment Derivatives

A number of microalgal pigments and compounds are listed in Table 3 [51,52,53,54,55,56,57,58,59,60,61,62]. Pigments like chlorophylls, carotenoids and phycobiliproteins are associated with health benefits observed in lab-based assays [63,64,65] and in vivo in animals [63,64,65,66], and when consumed by human subjects [67]. Astaxanthin, lutein, fucoxanthin, canthaxanthin, zeaxanthin and β-cryptoxanthin are used as antioxidants, anti-inflammatory agents, and as anti-tumor agents [68]. Benefits of carotenoids have also been shown using in vivo studies. Ranga Rao et al. [69] looked at bioavailability of *Spirulina platensis, Haematococcus pluvialis and Botryococcus braunii* biomass with a focus on β-carotene, astaxanthin and lutein. When the plasma, liver and eye tissue of rats following consumption of carotenoids were examined, astaxanthin and lutein were found in all tissues. *S. platensis*, *H. pluvialis* and *B. braunii* biomass could prevent lipid peroxidation through scavenging free radicals and hydroxy radicals. Some astaxanthin ester derivatives including astaxanthin mono and diesters obtained from the green algae *Haematococcus pluvialis* were also found to improve antitumor effects in rat [70,71]. 

Pigment production from algae varies when up-scaling from lab- to industrial-scaled production. Some active pigments including PBPs (blue pigment extracted from *Spirulina*), astaxanthin (yellow-to-red pigment extracted from *Haematococcus*) and β-carotene (yellow pigment extracted from *Dunaliella*) are produced at industrial scale with the end products used extensively [72]. The most extensive application for pigments is as food colorants. Pigments may also have antiseptic properties and may be used as preservatives due to their antioxidant activities, which may prevent spoiling of foods by inhibiting fatty acid oxidation. In addition, chlorophylls are known to be excellent deodorizers of foods [73]. 

Several factors can affect the cost of microalgal pigments extractions. These factors include the target algal organism, market trends and available technology. Extraction involves either non-mechanical methods like chemical, thermal, and enzymatic treatments or mechanical extraction using pressure and ultrasonics, microwave treatment or electric field treatment as well as supercritical fluid extraction (SCF). Yields of pigments depend on the cell wall structure of the alga and how well this can be disrupted as well as the solubility of the pigments in different solutes. Cell disruption methods include homogenization, CO_2_ SCF, omics heating and electric pulse field [74,75,76,77,78,79,80]. Pigment extracts and how they are produced are shown in Table 3. 

#### 2.2.2. Protein Derivatives

Protein value depends on protein content and size which results from refining, for example, achieved using filtration methods with size limits (e.g., 3 kDa or 10 kDa). Whole-cell protein contains 40–50% protein, protein concentrates contain 60–89% protein and isolates can contain between 90 and 95% proteins. Hydrolysates are usually 70–95% pure protein and bioactive peptides containing permeates have >95% protein purity [81]. Methods used to produce proteins result in its quality in terms of protein digestibility and functional activities. Processing can affect technofunctional attributes of proteins including emulsification and foaming properties [81,82,83,84], and different protein derivatives have different processing methods and applications. 

##### Protein Concentrates or Isolates

Concentrates and isolates find use in products like soup and sauces where the technofunctional attributes like absorption and emulsification properties are important as well as foaming and gelation [81]. Concentrates are also used in muffin manufacture, pastas and biscuits [85]. It was reported that emulsifying capacity and stability of microalgae concentrates are comparable to or even higher than ingredients like sodium caseinate [81,86]. The literature suggests different approaches for the production of protein concentrates but, in general, the following steps are followed [81]: microalgal harvest, spray drying, high-pressure homogenization (pH 8–10), clarification (centrifugation), membrane UF-DF (50–300 kDa MWCO) filtration and refining with enzymes (hydrolysis). 

Downstream processing needs to enable the recovery of functional proteins cost effectively and at high protein purity and yield. Disruption of algal cells using either high-pressure homogenization and milling at high pH values (pH 8–10) can be performed, followed by clarification with centrifuges or phase separation decanters. Additional purification is still required using filtration or protein precipitation. This approach can yield around 80% protein. Membrane filtration is more suitable than precipitation to produce protein concentrates due to the enhanced solubility and functionality of the proteins [86]. Additional purification steps result in isolate production—these have, on average, greater than 90% protein content. Protein precipitates and retentates that result from filtration include polysaccharides and small molecules [81,84].

##### Protein Hydrolysates and Bioactive Peptides

Protein hydrolysates are used in food and beverages where solubility and stability at elevated temperatures, which occur during processes like pasteurization, are present. Acid pH conditions are also suitable for protein hydrolysates. Enzymes and or acids are used in hydrolysis generation. Lipids are removed prior to hydrolysis usually or directly after hydrolysis using solvent systems or centrifugation. Ethanol is often used to extract high-value lipids and pigments prior to protein hydrolysis.

Bioactive peptides can be recovered from hydrolysates [81]. Proteinogenic amino acids and peptides from microalgae can be applied as antioxidants, antihypertensive agents, anticoagulants, and as anti-proliferative and immunostimulants [18,87]. Hydrolysate fractions less than 3 kDa in size usually contain bioactive peptides and these can be recovered using different filtration methods including tangential flow filtration and MWCO methods. 

#### 2.2.3. Lipid Derivatives

Microalgae are promising sources of natural edible oils for nutritional application in foods [88,89,90] as they contain PUFAs and produce oils efficiently compared to land-based crops. Microalgae make lipids in the triacylglycerol (TAG) form and these lipids can be used in products like infant formula as an alternative to human milk TAGs [91]. Microalgae increase lipid production when stressed, for example, under conditions of intense light or high salinity as well as under a combination of stressful conditions [92]. 

PUFAs which include the n-3 and n-6 classes are known for their health benefits [93] and have beneficial properties for the cardiovascular system, anti-cholesterol activity, and other bioactivities [94,95]. Omega-3 fatty acids include alpha-linolenic acid (ALA) (18:3, n-3), stearidonic acid (STA) (18:4, n-3), eicosapentaenoic acid (EPA) (20:5, n-3), docosapentaenoic acid (DPA) (20:5, n-3) and docosahexaenoic acid (DHA) (22:6, n-3). Omega-6 fatty acids include linoleic (LA, C18:2), ɣ-linolenic (GLA, C18:3) and arachidonic (ARA, C20:4) acid. PUFAs are used in nutraceuticals, pharmaceutical and therapeutic applications [96]. EPA and DHA are known for their health benefits and the market for these is expansive [97]. PUFA production for use in infant formula is a significant sector and includes DHA [98]. GLA and ARA from species like *Arthrospira platensis*, *Porphyridium cruentum*, *Mortieriella alpine* and *Parietochloris incisa* find use in supplements. Some microalgal sources of lipids and the potential applications of algal-derived PUFAs are shown in Table 4. 

#### 2.2.4. Carbohydrate Derivatives

Microalgal polysaccharides are stable, and versatile and generally regarded as safe [99]. Microalgal polysaccharides consist of the sugars galactose, xylose, and glucose. Polysaccharides from microalgae consist usually of β-glucans, cellulose, hemicellulose and uronic acids as well as fucose [100]. Exopolysaccharides are known to be very bioactive with activities that include antioxidant, anti-inflammatory and antimicrobial activities [99]. Several factors including growth conditions of the microalga affect the polysaccharide concentration [101]. The use of microalgal polysaccharides was reported previously and they find application in functional foods, nutraceuticals, and supplements, namely as sources of dietary fiber [102], as food thickeners [103] or food ingredients for weight management [104]. 

Microalgal sources of polysaccharides include *Porphyridium* sp., *Chlorella* sp., and *Spirulina* sp. [101]. *S. platensis* polysaccharides can be extracted using enzyme hydrolysis, ultrasound and other pre-treatments, and are known to be very nutritious [105]. *Chlorella pyrenoidosa* and *Spirulina platensis* polysaccharides have known anti-obesity properties [106] and those isolated from *Chlorella vulgaris* can prevent airway inflammation [107]. 

Polysaccharides are extracted usually with centrifugation and microfiltration to remove cells from polysaccharides. The solvents methanol, ethanol, isopropanol, or acetone can be applied for precipitation. Sonication and other physical methods or chemicals like formaldehyde, hot water, or sodium hydroxide or resins with ionic capacity can help recover the polysaccharide fractions [108,109].

### 2.3. Legislation Concerning Microalgae Use as Food in EU and USA

#### 2.3.1. EU

EU regulations apply to microalgal use as foods in the EU, as shown in Table 5. Food safety legislation is also of great importance for microalgal use as foods or feeds. The history of use as food of the alga is important for its regulatory status. The Novel Food Regulation which states that “species having not been used as food to a significant degree in any of the EU member countries before 15 May 1997 need to undergo authorization procedures in order to ensure their safety for human consumption (Regulation (EC) No 258/97)”.

Table 6 cites relevant regulations that are relevant to food and feed development and use in the EU. The New Novel Food Regulation (EC) 2015/2283 provides for species that have a demonstrated history of safe use (equal to or greater than 25 years) external to the EU.

The novel food catalogue contains the EU list of all approved novel foods. The catalogue collates both imported and EU algae, and there are around 22 approved algae listed here (https://ec.europa.eu/food/safety/novel-food/novel-food-catalogue_en (accessed on 11 January 2022)) including *Arthrospira platensis*, *Chlorella luteoviridis*, *Chlorella pyrenoidosa*, *Chlorella vulgaris*, *Chlamydomonas reinhardtii*, and *Spirulina* sp.

Regulation EC/1924/2006 concerns nutrition and health claims. EFSA evaluates the scientific evidence supporting these claims. Annex XIII details the recommended daily allowances (RDA) for foods (Nutrition Information Regulation (EU) 1169/2011). Iodine has an RDA of 150 µg. Algae are the only non-animal source of omega-3 PUFAs—EPA and DHA specifically. If a product contains 0.3 g ALA per 100 g and per 100 kcal, 40 mg of EPA and DHA per 100 g and per 100 kcal, a claim like “high in omega-3 fatty acids” can be made on a product according to Regulation (EC) 1924/2006.

Microalgae with a high contaminant level cannot be put on the market. Substances of relevance are dioxins, aflatoxins, heavy metals (such as lead and mercury) and nitrates. Microalgae may accumulate heavy metals from surrounding waters/wastewaters. Heavy metals need to be assessed. Commission Regulation (EC) No 1881/2006 on contaminants sets permitted levels of heavy metals for food. There is no maximum level of cadmium or inorganic arsenic set for microalgae. Regulation (EC) No 396/2005 sets the permitted maximum level for mercury at 0.01 mg/kg. The maximum limit for cadmium is 3.0 mg/kg for food supplements. The maximum limit for mercury and lead are 0.1 mg/kg and 3.0 mg/kg. EU Regulation (EC) No 1333/2008 governs food additive use in the EU, and contains eight algal additives (codes E401-E407a). 

Foodstuffs may contain pathogens that pose a food safety risk. Good Hygiene and Manufacturing Practices (GHP, GMP) and Hazard Analysis Critical Control Point (HACCP) principles should be followed where algae are used as foods to ensure food safety. Commission Regulation (EC) No 2073/2005 concerns microbiological criteria for foods (January 2006). 

#### 2.3.2. USA

Generally Recognized as Safe (GRAS) is granted by the FDA to substances considered safe for human consumption. GRAS can be obtained by an operator by use of documented evidence of human consumption. Alternatively, it can be determined by substantiating scientifically the safety of the substance. The purity of a substance is important to ensure safety. Securing GRAS status requires time and money, and currently only microalgal species *Spirulina* sp., *Chlorella* sp., *Dunaliella* sp., *Haematococcus* sp., *Schizochytrium* sp., *Porphyridium cruentum* and *Crypthecodinium cohnii* have GRAS status. Oil from *Schizochytrium* and *Ulkenia*, as well as a whole microalgal protein powder and a lipid ingredient derived from *Chlorella* sp. also have GRAS status in the USA.

## 3. Current Market Scenario for Microalgal Use as Feed

### 3.1. Current Market Scenario

The utilization of microalgae as livestock feed greatly depends on the type of microalgae and their nutrient composition, as well as animal adaption to the ingredient [33]. Research demonstrates that utilizing microalgae biomass in animal feed could enhance the immune response, durability towards illness as well as antibacterial and antiviral action. *Spirulina* sp. and *Chlorella* sp. are permitted for feed use only currently. *Dunaliella* sp. for pure β-carotene is permitted for use, and production of astaxanthin from *Haematococcus* sp. is allowed as a feed additive. *Chlamydomonas* sp., *Chlorococcum* sp., and *Scenedesmus* sp. are allowed in aquaculture feed but do not have GRAS status. 

#### 3.1.1. Aquaculture

Fishmeal use in aquaculture is considered not sustainable and microalgae are growing in popularity in this application field as they are protein- and oil-rich [121]. Species including *Chlorella*, *Isochrysis*, *Pavlova*, *Phaeodactylum*, *Chaetoceros* and *Nannochloropsis* are often used in fish feed [122]. Table 7 shows the proximate composition of algae compared to fishmeal and soy. In addition, the amino acid profile and high levels of PUFAs also make microalgal use in feed and nutrition extremely effective [123].

Microalgae can improve stress responses in fish as well as growth. They positively impact against disease development in fish and improve carcass quality [126]. Pigment in fish flesh is also increased by consumption of microalgae likely due to carotenoid content of the microalgae used as feed [121]. Table 8 shows examples of commercially available microalgae used as feed ingredients and their observed health benefits. 

#### 3.1.2. Livestock and Poultry Feed

Microalgal inclusion in livestock feed can improve weight gain, health and end product quality, as shown in trials with rabbits and poultry previously [132,133]. *Arthrospira* sp. was applied in the feed of pet animals—dogs, cats, and ornamental birds previously as well as in the diet of pigs, horses and cattle. *Arthrospira platensis* increased average daily weight gain in pigs in a trial. *Schizochytrium* sp. was found to improve the fatty acid composition of pork and poultry. When used to feed chickens, microalgae improve the yellow color of egg yolks [133]. *Chlorella* sp. was found to improve poultry growth performance. These examples illustrate that microalgae, as feed ingredients, are a promising alternative to corn and soybean. Table 9 shows the benefits of microalgal inclusion in feeds on animal health and growth.

### 3.2. Legislation Concerning Microalgae Use as Feed in EU and USA

#### 3.2.1. EU

Two regulations relate to feed on the EU market—Regulation EC No 767/2009 related to putting feed on the EU market [141,142] and Regulation EC No 183/2005 concerning feed hygiene [143]. The goal of both regulations is feed safety. An animal diet/feed ingredient must be on the list of permitted items on the ‘Feed Materials Register’ (www.feedmaterialsregister.eu (accessed on 11 January 2022)). If not on this list, the material has to be announced via notification (art. 24 (part 6), Council regulation 767/2009/EC). Feed safety is regulated by Council Regulation 767/2009/EC (e.g., Art. 4), the General Foodstuff Council regulation (e.g., Art. 15), and the hygiene regulations in Council Regulation 183/2005/EC. Feed additives are regulated by Council Regulation 1831/2003 [144] and there is a European Union Register of Feed Additives. Until 2013, microalgae as animal feed additives were not allowed due to concerns regarding manure and digestate use [145]. The GMP+ legislation was changed and microalgae can be used as dietary feed ingredients for animals following a risk assessment. Accepted feed materials in the EU, including algal meals as well as algae oil and extracts are defined in Commission Regulation 68/2013. Toxic contaminants in animal feed are detailed in Directive 2002/32/EC. Arsenic is permitted in feed at a concentration of 10 mg/kg in complete and complementary feed for pet animals and 40 mg/kg for algal meal and algae derived feed materials (Directive 2002/32/EC). Inorganic arsenic must be lower than 2 mg/kg. Amounts of 10 mg/Kg, 0.1 mg/kg and 0.1 mg/kg are the maximum permitted levels of lead, cadmium and mercury in feeds. The increase in the global population has raised questions about what is fed to companion animals and pets and the need for more sustainable ingredients. The nutritional composition and metabolomic profile of the microalgae *Tetradesmus obliquus*, *Chlorella vulgaris*, and *Nannochloropsis oceanica* met the requirement for essential amino acids (except for cysteine and methionine) in the diet of dogs at all stages of life. Additionally, microalgae provided the FEDIAF-approved amount of fatty acids required in the diet of dogs. FEDIAF regulates and provides guidance on pet ingredients and their use in the EU [146].

#### 3.2.2. USA

The FDA Center for Veterinary Medicine (CVM) are authorities on feed in the USA. The Association of American Feed Control Officials (AAFCO) publishes feed ingredient definitions in the AAFCO official publication. Laws concerning feed in the US that permit use of algae as feeds include the Federal Food, Drug and Cosmetic Act (FD&C) which regulates all food and feed additives introduced since 1938 and the Dietary Supplement Health and Education Act (1994) that includes the dietary supplement sectors. 

## 4. Applications of Microalgae Produced on Wastewaters

### 4.1. Benefits of Cultivating Microalgae on Wastewaters

Growth of microalgae on wastewater allows for a reduction in nitrogen and phosphate, and a decrease in both biological and chemical oxygen demands [147]. Organic wastes are carbon-rich and microalgae have grown on them successfully, previously [148,149]. To remove carbon, nitrogen and phosphorus from waste waters, different algae have been used and their growth assessed on agricultural, brewery, and industrial effluents, with promising results in terms of growth and nutrient removal [150,151,152,153]. *Scenedesmus obliquus* removed ammonia and nitrogen from municipal wastewater effectively previously [150]. *Chlorella* sp. grown on brewery wastewater also was found to remove 94.38% ammonia and 88.52% nitrogen with a decrease in COD [152]. 

Microalgae can use urea (organic N) and ammonium (inorganic N), and nitrates [154]. Ammonium is the preferred N source for microalgae as it does not require reduction steps in algal cells [155]. Using ammonia-containing wastewaters for microalgal growth may help to recover nitrogen. This is positive in terms of Greenhouse gas (GHG) emissions associated with microalgal culture and several studies report the negligible emission of N_2_O caused by microalgae in wastewater treatment [156,157]. Alcántara [158] established a microalgae wastewater treatment process. This was estimated to have an emission factor of 0.0047% g N_2_O-N g^−1^ N-input. 

Microalgae can remove heavy metals and chemicals from wastewaters [159]. Pharmaceutical manufacture results in chemicals in the water course. It has been reported that over 200 substances are released into water bodies, including antibiotics like ciprofloxacin from Pharma [160]. *Nannochloropsis* sp. was shown to remove acetaminophen (commonly known as paracetamol), ibuprofen and olanzapine from industry wastewater [161]. In addition to pharmaceuticals, microalgae can absorb heavy metals. *Chlorella miniata* and *Scenedesmus quadricauda* were used to remove zinc and nickel from wastewater [162]. *Scenedesmus quadricauda* reduced nickel and zinc by 99%, *Chlorella miniata* reduced the amount of nickel by 70%. Additionally, Bellucci et al. [163] demonstrated the removal of *Escherichia coli* from waste water previously. 

To produce 100 tons of microalgae biomass, up to 10 tons of N and 1 ton of P are needed [164]. Wastewater treatment and microalgal cultivation could concurrently reduce nutrient costs for algal production and clean up water courses in a more economic manner [13,165]. In short, utilizing microalgae with wastewater treatment effectively can clean wastewaters and prevent eutrophication. Microalgae could be used for energy and fuel production when produced on wastewaters [166,167] but the safety of this process must be assessed firstly. 

### 4.2. Caveats of Cultivating Microalgae on Wastewaters

Limits and challenges of cultivating microalgae with wastewater also have to be considered. Limitations include the following:(i)Microalgal bioremediation is seasonal and may not be suitable for use in the winter when UV levels are low and microalgal growth is hampered by UV and temperatures.(ii)CO_2_ supply is critical for microalgal growth and therefore wastewater treatment [168]. CO_2_ sources near the cultivation site are needed.(iii)Very high concentrations of contaminants have a negative impact on microalgae. An adaptation step for the microalgal strain to the target compound(s) is required on site.(iv)High concentrations of ammonia are toxic to many microalgal strains and pH control in wastewaters is needed to ensure ammonia concentrations are regulated [154]. Low concentrations of nitrogen and/or phosphorus in wastewater result in little microalgal biomass and costs where this happens are high and the process could be nonviable [169].(v)Xenobiotic pollutants require a combination of microalgae for their removal. Consistency in polyculture biomass quality can be a concern.(vi)The effect of annual variations in light and temperature on the bioremediation efficiency and microalgal biomass quality and yield needs further work as production of algae at a large-scale is currently limited.(vii)Additional applications of wastewater-grown microalgal biomass are required to make production processes economically viable in the context of the bio-economy.

### 4.3. Current Applications of Microalgae Produced Using Wastewaters

Microalgae cultivated on wastewaters cannot be used for human applications. Algae produced in this manner find application as biofertilizers, biofuels and bioplastics, which do not directly affect the human food chain. It is not permitted to use algal biomass recovered from urban wastewater in animal feeds except for manure application [170]. However, microalgae produced on waste streams may have potential for use as animal feed ingredients if the safety and quality of the end algal product is achieved through processing. Strict control of toxic bacteria and microbiological assessment for pathogens in wastewater treatment is necessary and important to determine biomass safety [171,172], as bacteria which could contain toxic compounds or release bio-toxin co-exist with microalgae especially growing in wastewater, and pose a threat to the food chain. Microalgae produced on “waste” streams and their potential applications are in Table 10. The microalgae used in these studies showed not only high nutrient recovery ability, recovering nitrogen, phosphorus, and reducing chemical oxygen demand, but also positively impacted animal growth performance in several instances. 

### 4.4. Legislation Concerning Use of Microalgae Grown on Wastewater as Biostimulants and Fertilizers

Other applications of microalgae grown on wastewaters include use as biostimulants or fertilizers. Regulation EU2019/1009 dictates the permitted levels of cadmium, hexavalent chromium, mercury, arsenic, nickel and other heavy metals in fertilisers. An amount of 2 mg/kg cadmium is permitted in sewage sludge fertiliser and this is proposed to be reduced to 0.8 mg/kg by 2030 in some EU countries (Sweden, for example). The maximum level for cadmium in bio fertilisers in the EU is 1.5 mg/kg (Regulation EU 2019/1009). Under the above regulation, a list of EU-accepted biostimulants would be created. Maximum limits for heavy metals are identical to those for fertilisers. Only scientifically proven claims can be put on a label concerning biostimulant action. Ricci [146] describes principles for justifying plant biostimulant claims and field trials needed. CEN is also developing a standard on this.

## 5. Conclusions

The ability of microalgae to make bioactive components makes it a promising raw material for many applications in food, feed and biostimulant uses. This review detailed potential uses of microalgae in food and feeds as well as caveats concerning their use. Current applications of microalgae are summarized including the compounds of interests with their current and potential applications, along with the processing methods for their production and the legislation concerning their use in the EU and USA. However, with current downstream processing techniques, multiple-product extraction from microalgae is not economically viable. This issue can be tackled by combining wastewater treatment and microalgal cultivation to reduce nutrient costs. However, many challenges have to be considered when considering the potential applications of microalgae grown on wastewaters, specifically the safety of the resulting biomass as well as consumer perception and public acceptability. Hence, in-depth investigations and further research are required in this field.

## Figures and Tables

**Table 1 foods-12-03878-t001:** Nutraceutical products made using microalgae or components of microalgae and delivery in different food and supplement forms.

Bioactive Ingredient	Microalgal Source	Claimed Health Benefit	Examples of Products (Trade-Name)	Producing Company
Carotenoids	*Dunaliella salina*	Antioxidant activity	Supplements—β-carotene patented Betatene^®^	Cyanotech, Mera Pharmaceuticals, AstraReal AB, Jingzhou Naturals
Astaxanthin	*Haematococcus pluvialis*	Immunomodulation	Supplements—Max botanics astaxanthin supplements/sunscreens	Mera Pharmaceticuals, AstraReal AB, Jingzhou Naturals Astaxanthin Inc. Max Botanics UK and Europe.
Beta-carotene	*Chlorella vulgais*	Retinal degradation prevention	Supplements—Dr. Mercola, Fermented Chlorella with Chlorophyll, 450 Tablets	Mercola, Florida
Lutein	*Chlorella pyrenoidosa*	Muscular dystrophy prevention	Terranova freeze dried *Chlorella pyrenoidosa*	Terranova Synergenistic nutrition, UK
Cantaxanthin	*Chlorella ellipsoidea*	Helps to prevent cancer; This carotenoid can also positively impact human health and enhance the appeal of foods with egg.	Animal feed ingredient—Used in animal feed for chickens to increase shelf life of eggs, health benefits for chickens	BASF SE, Wellgreen Technology Co., Ltd., and Nikken Shohonsha Co., Japan are producers of Cantaxanthin.
Violaxanthin	*Dunaliella tertiolecta/Chlorella ellipsoidea*	Anti-proliferative, anti-inflammatory, and proapoptotic activity against human cancer cell lines in vitro	Supplements	Cognis Nutrition and Health, Illnois, United States
Sulphated polysaccharides	*Porphyridium* sp., *Rhodella reticulata*	Antioxidant, Anti-aging, used in cosmetics as a substitute for hyaluronic acid, antibacterial, anti-inflammatory	Acqualift^®^—dried powder supplied in air and light sensitive packaging	NaturZell—BASM blue International
β-1-3-glucan	*Skeletonema* sp., *Porphyridium* sp., *Nostoc flegelliforme; Euglena gracilis*	Actives receptors on white blood cells and activates the immune system following the binding to the beta glucans.	BioGlena™—a next generation source of Beta-glucan	Algatech Ltd., Southern Israel, produces β-1,3-glucan from *Euglena gracilis*
Bioecolians is a a-gluco-oligosaccharide, composed out of short glucose chains linked by glycosidic bond(α1-2) and (α1-6).	Unknown	Gut health, prebiotic action	Bioecolians^TM^	Solabia-Algatech Nutrition, Israel and France
Docosahexaenoic acid (DHA)	*Schizochytrium* sp.	Emulsions for beverage fortification	Source Oil^®^/Algae Omega-3	Unknown
Algal inks	*Arthrospira platensis/Chlorella vulgaris*	3D printed cookies/edible inks/used in dying clothes	Fristads (producers of algal inks for dying garments—start up)	Fristads, Sweden
*Chlorella vulgaris* protein and Vitamin B12, whole protein (biomass)	*Chlorella* sp.	Ice cream, milk, cheese	Algal proteins or whole biomass	Sophie’s BioNutrients, Singapore, Solazyme who became erravia and were taken over by Corbion, A Dutch company. Cheese was a collaboration with Ingredion, Singapore
Algal protein ingredients	*Chlamydomonas* (Hardtii) sp.	Algal protein ingredients and seafood alternatives	Hardtii Alt-Meat and Tuna products	Triton Algae Innovations, San Diego, USA

**Table 2 foods-12-03878-t002:** Microalgae relevant for food/feed applications and their safety aspects.

Species	Safety Aspect	Species	Safety Aspect
*Arthrospira platensis*	GRAS ^1^	*Navicula* sp.	NT ^2^
*Synechococcus* sp.	NT	*Nitzschia dissipata*	NT
*Tetraselmis* sp.	NT	*Phaeodactylum tricornutum*	NT
*Chlamydomonas reinhardtii*	NT	*Thalassiosira pseudonana*	NT
*Haematococcus pluvialis*	GRAS	*Odontella aurita*	NT
*Dunaliella* *bardawil*	GRAS	*Skeletonema* sp.	NT
*Chlorococcum* sp.	NT	*Monodus subterraneus*	NT
*Scenedesmus*	NT	*Nannochloropsis* sp.	NT
*Desmodesmus* sp.	NT	*Isochrysis* sp.	NT
*Chlorella protothecoides*	GRAS	*Pavlova* sp.	NT
*Parietochloris incisa*	NT	*Crypthecodinium cohnii*	GRAS
*Porphyridium cruentum*	GRAS		

This table is cited from Lucakova et al. [48]. ^1^ GRAS refers to Generally Recognized as Safe. ^2^ NT refers to no toxins known.

**Table 3 foods-12-03878-t003:** Use of algae in food products, different food delivery formats and observed health benefits.

Microalgae Species	Bioactive Components	Food Product	Sensory Effect	Form	Health Benefits	Reference
*Chlorella sorokiniana*	Protein, chlorophyll, and carotenoids	Pasta	Improved colour	N.A. ^1^	Prevention of foodborne diseases	[51]
*Chlorella* sp.	Protein, PUFA ^2^ -ω3, EPA ^3^, DHA ^4^	Milk	Improved flavour and mouthfeel	Powder or liquid	Reduced risk of anaemia	[52]
*Chlorella vulgaris*	Protein, chlorophyll pigment	Cookies	Improved colour stability	Powder	High antioxidant activity	[53]
*Chlorella vulgaris*	Protein, pigments (chlorophylls, phycocyanin and canthaxanthin)	Pasta	Improved colour and texture	Powder	Prevention of gastric ulcers, constipation and anaemia	[54]
*Scenedesmus obliquus*	Protein, PUFA, EAA	Functional chocolate	No significant difference in the texture	N.A.	Prevention of cardiovascular diseases, hypertension and inflammation	[55]
*Nannochloropsis* sp.	EPA, phenolic compounds	Pasta	Improved appearance and colour	N.A.	High antioxidant activity	[56]
*Nannochloropsis* sp.	Phenolic compounds, bioactive peptides, carotenoids	Functional breads and crackers	Improved nutritional value	N.A.	Increased antioxidant capacity	[57]
*Spirulina* sp.	Proteins, lipids and carotenoids	Ready-to-eat snacks	High sensory acceptance index (82%)	Powder	Increased nutritional composition	[58]
*Spirulina* sp.	Phenolic compounds	Bread wheat pasta	N.A.	N.A.	Increased antioxidant activity	[59]
*Dunaliella salina*	Protein, fatty acids, phenolic compounds	Functional cookies	Good mouthfeel, colour and appearance.	Powder	Increased moisture and antioxidant activity	[60]
*Schizochytrium* sp.	PUFA	Dry-fermented meatproducts	No significant difference in appearance and odour	Pre-emulsified algae oil	Showed better omega-6 to omega-3 ratio and increased DHA content, increased stability to the oxidation	[61]
*Isochrysis galbana* *Nannochloropsis oculata*	Chlorophyll-a and carotenoid	Chewing gum	No significant difference in appearance, chewiness, adhesiveness and alga taste of *I. galbana.*	Dried	Increased pigment value	[62]

^1^ N.A. refers to not available. ^2^ PUFA refers to polyunsaturated fatty acid. ^3^ EPA refers to Eicosapentaenoic acid. ^4^ DHA refers to Docosahexaenoic acid.

**Table 4 foods-12-03878-t004:** Microalgal pigments: extraction and applications.

Microalgae	Pigment	Extraction Method	Application	References
*Chlorella sorokiniana*	Lutein	Use of potassium hydroxide (KOH) and L-ascorbic acid combined with microwave	Industrial applications for marine microalgae	[75]
*Scenedesmus almeriensis*	Lutein	Ball milling; extracted with ethanol	Commercial production of lutein	[76]
*Haematococcus pluvialis*	Astaxanthin	Ultrasonicated; extracted using ethyl acetate	Pharmaceutical (encapsulation)	[77]
*Arthrospira platensis* IFRPD 1182	phycobiliproteins	Ultrasonication	Food addictive	[78]
*Nannochloropsis gaditana*	β-carotene	Ultrasonication combined with solvents like acetone following freeze-drying	Food and nutraceutical	[79]
*Synechococcus elongatus*	Zeaxanthin	chemical products for food, pharma use	Chemical products for food and pharmaceuticals	[80]

**Table 5 foods-12-03878-t005:** Microalgae fatty acids and potential application.

Fatty AcidFraction	Microalgae Source	Applications	Daily Intake Recommendation for Human (mg)	References
Omega-3				
Eicosapentaenoic acid (EPA)	*Nannochloropsis* *oculata*	Brain development for children, cardiovascular health	250–500	[74,110,111,112]
	*Phaeodactylum* *tricornutum*			
	*Monodus* *subterraneus*			
	*Isochrysis galbana*			
	*Pavlova lutheri*			
Docosahexaenoic acid (DHA)	*Pavlova lutheri*	Food supplement, infant formulas for full-term/preterm infants, significant for cardiovascular, supplements	250–500	[74,96,113,114,115,116]
	*Schizochytrium* *limacinum*			
	*Crypthecodinium cohnii*			
Alpha-linolenic acid (ALA)	*Chlorella vulgaris*	Nutritional supplement	1000–2000	[117,118]
	*Chlamydomonas reinhardtii*			
Omega-6				
Gamma-linolenic acid (GLA)	*Arthrospira* *platensis*	Inflammation prevention, auto-immune diseases	500–750	[119,120]
Arachidonic acid (ARA)	*Porphyridium* *cruentum*	Anti-inflammatory, muscle anabolic formulations	50–250	[115,118]
	*Mortieriella alpina*			
	*Parietochloris incisa*			

**Table 6 foods-12-03878-t006:** Key EU regulations concerning microalgae for use as foods/food ingredients.

Article	Date of Issue	Content	Aim	Critical Issues
**Novel foods and novel food ingredients**				
EC 258/97	15 May 1997	Set out the legal framework for the marketing of “Novel Foods” and provide a system of authorization for the novel food marketing	To grant a high level of consumer protection and the functioning of the internal market.	Takes 3 years, high costs for novel food status (>€200,000).
EU 2015/2283	25 November 2015	Algae/extract considered to be novel food if it has not been consumed to a significant degree within the Union before May 15, 1997	To protect consumers.	Complex frameworks of policies to progress the microalgae food industry are challenging and time-consuming.
EU 2017/2407	20 December 2017	Maintained an online list called the novel food catalogue that contains the unions list of all authorized novel foods. The novel food catalogue contains both European and imported algae.	To summarize the novel food catalogue up to date (https://food.ec.europa.eu/safety/novel-food/novel-food-catalogue_en (accessed on 11 January 2022)).	
**Food safety**				
EC 2002/178	28 January 2002	Provided EC (2002/178) provided a framework for a coherent approach in the development of any food legislation.	To ensure food safety during food production and distribution.	Relates to commonly consumed foods, not “new” foods.
**Nutrition and health**				
EC 1924/2006	20 December 2006	Health claims should be supported with science and substantiated.	Scientific substantiation applied when making the claims.	

**Table 7 foods-12-03878-t007:** Typical composition of commercially available feed ingredients and microalgal species.

Feed Ingredients/Microalgae	Protein (% DW)	Lipid (% DW)	Carbohydrate (% DW)	Ash (% DW)	References
Fish meal	63.0	11.0	N.A ^1^	16.8	[124]
Soy bean meal	44.0	2.2	39.0	6.1	
Corn-gluten meal	62.0	5.0	18.5	4.8	
Wheat meal	12.2	2.9	69.0	1.6	
*Chlorella* sp.	52.0	7.5	24.3	8.2	
*Chlorella pyrenoidosa*	57	2	26	N.A.	[123]
*Chlorella vulgaris*	51–58	14–22	12–17	N.A.	
*Chlamydomonas rheinhardii*	43–56	14–22	2.9–17	N.A.	
*Scenedesmus obliqus*	50–56	12–14	10–52	N.A.	
*Nannochloropsis granulata* (CCMP-535)	33.5	23.6	36.2	6.7	[125]

^1^ N.A. refers to not available.

**Table 8 foods-12-03878-t008:** Commercially available microalgae use as aquaculture ingredients and health benefits.

Microalga	Fish Trial	Feed Product	Suggested Content	Components of Interest	Health Benefits	Reference
*Chlorella* sp.	Nile tilapia (*Oreochromis niloticus*)	Fish meal substitution	Up to 50%	N.A.	Enhanced fish growth, feed conversion ratio and protein-productive value	[127]
*Spirulina* sp.						
*Chlorella ellipsoidea*	Juvenile flounder (*Paralichthys olivaceus*)	Dietary supplementation	2%	N.A.	Enhanced growth, feed utilization, serum cholesterol level, and whole-body fat contents	[128]
*Spirulina platensis*	Great sturgeon (Huso huso Linnaeus, 1754)	Dietary supplementation	10%	N.A.	Enhanced growth and activation of immune responses	[129]
*Scenedesmus almeriensis as*	Juvenile gilthead sea bream (*Sparus aurata*)	Fishmeal alternative	20%	Protein	Increased the level of intestinal enzyme activities and intestinal absorptive surface	[130]
*Scenedesmus* sp.	Atlantic salmon (Salmo salar L.)	Fishmeal diet	Up to 10%	EAA, PUFA	Improved the total n-3 and PUFA content in salmon	[131]

**Table 9 foods-12-03878-t009:** Benefits of microalgae use in feeds.

Microalgae	Level in the Diet (% DM)	Experiment Duration	Animal	Main Findings	References
*Arthrospira platensis*	0.01%	5 weeks	Lamb	Increase in ADG ^1^, ADFI ^2^ and final body weight; FCR negatively impacted ^3^	[134]
*Arthrospira platensis*	10–20%	6 weeks	Weaned lamb	No negative impacts observedBody weight positively impacted (increase) (10% dosage)	[135]
*Chlorella vulgaris*	0.1–0.2%	6 weeks	Female Pig	Increase in ADG (0.1% dosage); no negative impacts observed	[136]
*Chlorella vulgaris*	0.07–0.21%	42 days	Day-old broiler chicks	No negative effects observed	[137,138]
*Chlorella* sp.	0.1–0.2%	42 days	Day-old male Pekin ducks	Increased feed intake	
*Porphyridium* sp.	5–10%	10 days	Thirty-week-old chickens	Negative, reduced ADFI	[139]
*Arthrospira platensis*	5–15%	24 days	Weaned rabbits	Increase in ADFI (10% dosage)	[140]

^1^ ADG refers to average daily gain. ^2^ ADFI refers to average daily feed intake. ^3^ FCR refers to feed conversion ratio.

**Table 10 foods-12-03878-t010:** Potential applications of microalgae cultivated in different waste streams used as animal feed.

Microalgae	Cultivation Medium	Cultivation Mode	Production/Productivity	Nutrient Removal	Final Product	Target Compound	Potential Commercial Development	Health Benefits	Further Analysis	References
Microalgae consortium (*Chlorella* sp. *Scenedesmus* sp.)	Anaerobic digested piggery effluent	Semi-continuous	2.20 ± 0.49 gm^−2^d^−1^ (*n* = 5)	N-NH_4_: 1.97 ± 0.32 gm^−2^ d^−1^ (*n =* 5) COD: 5.83 ± 1.37 gm^−2^ d^−1^ (*n* = 5)	Pig feedstock meal	Protein; PUFA	Replacement for soybean meal	ω−3:ω−6 ~1.9 value-adding use	In vivo studies and long term feeding trials	[173]
*Chlorella vulgaris*	Seafood Production effluent from crabs	Semi-continuous	1.1 Kg (DW)/week	TN: 100%; TP: 96.5%; COD: 96.2%; BOD_5_: 99.7%	Fish diet supplement	Protein; carbohydrates	Process as pellets for aquaculture fish feed	High protein; Carotenoid content for attracting colours	Optimal inclusion for different aquaculture species diet	[174]
*Scenedesmus obliquus*			0.82 Kg (DW)/week	TN: 100%; TP: 98.6%; COD: 97.7%; BOD_5_: 99.7%						
*Scenedesmus* sp.	Chicken slaughterhouse wastewater	Batch	/	/	Fish feeds	Carbohydrate; protein; lipids—EPA	Fish feeds	High EPA content	/	[175]
Microalgae consortium	90% agricultural runoff + 10% domestic wastewater	Continuous pilot-scale	/	Se: 43%; N-NH_4_: 93%; TP: 77%; TCOD: 70%	Feed supplements	Protein; PUFA	Se-enriched soybean replacement	High protein content, ω−3 and ω−6 content	Risk assessment and downstream processing after harvest to reduce bacteria loads and pathogen risk	[176]
*Haematococcus pluvialis*	Synthetic dairy wastewater	/	0.55 ± 0.01 g L^−1^	TN: 79–81%; TP: 57–79% and TCOD: 94–96%	Feed stock	Astaxanthin, lipids, and carbohydrates	High value feed stock	High astaxanthin production	Optimization of cultivation conditions	[177]
*Chromochloris zofingiensis*			0.65 ± 0.01 g L^−1^							
*Nannochloropsis gaditana*	Fish farm effluents	Batch	847.0 mg L^−1^	/	Fish feeds	PUFA	Fish feed or prey of zooplankton	High protein and polysaccharide content	/	[178]
*Nannochloropsis salina*	Pre-gasified industrial process water	Batch	/	/	Aquaculture feed	EPA	Fish feed rich in EPA, protein, tocopherols and carotenoids	High EPA, protein, tocopherols and carotenoids content	Further extraction and fractionation of the biomass	[179]

## Data Availability

Not applicable.

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
