# Peer review of "Applications of Microalgae in Foods, Pharma and Feeds and Their Use as Fertilizers and Biostimulants: Legislation and Regulatory Aspects for Consideration"

_foods, 2023, doi:10.3390/foods12203878_

Round 1
Reviewer 1 Report
Comments and Suggestions for Authors
Overall the manuscript has been written well. However, there are some comments :
1) Section 2.1 - Current market Scenario
- I recommend to add some discussion/information/examples on the nutraceutical product.
- The nutraceutical examples can also been added in Table 1
2) Section 2.3.1 EU
Line 369 - 378 : Recommended the information on the maximum limit of the contaminants are more appropriate to be listed in a Table.
3) Section 4.3 Current applications of microalgae produced using wastewaters
- Separate the explanation on legislation of fertilizer to different subsection
Author Response
Please see attached the response to reviewer comments one. We have adjusted the paper accordingly.
Kind regards,
Maria

Reviewer 2 Report
Comments and Suggestions for Authors
In my opinion, the manuscript meets all the requirements. The topic is interesting, the purpose of the research is well set. The literature review was well done. The large number of tables allows easy navigation among the most relevant information. The discussion is well described. List of references prepared very carefully. The authors used as many as 179 literature items. I recommend it.
Author Response
Please see attached the response to reviewer comments 2 - we would like to thank the reviewer for their time in reviewing our manuscript.

Reviewer 3 Report
Comments and Suggestions for Authors
This paper collates an overview of existing algal applications and current market scenarios for microalgal products as foods and feeds along with of relevant legislative requirements concerning their use. This is a good work and well written. Below are some suggestions.
1. This in an interesting work and minor revision is recommended. The title is not specific enough. This paper is mainly about food and feed, and it is recommended to modify the title according to the main content.
2. It is suggested to supplement some literature in introduction part, analyze the differences between this paper and previous literature, and further emphasize the novelty of this paper.
3. Introduction. Moreover, they are known for their potential use as biofuels in renewable energy, as well as ingredients for use in biostimulants, cosmetic compounds, and healthcare products. Microalgae is an important raw material of biofuel, which has attracted more and more attention. Some recent literature can be referenced to improve the quality of this work, such as "Lipid accumulation by a novel microalga Parachlorella kessleri R-3 with wide pH tolerance for promising biodiesel production".
4. Table 1. "sp." should be shown in italic form. Please check the whole manuscript.
5. It is suggested to further analyze the limitations faced when using microalgae as food or feed, as well as how to carry out industrial production.
6. Section 4 Applications of microalgae produced on wastewaters. Tthe reduction of organic substrates by decreasing parameters such as biological oxygen demand (BOD) and chemical oxygen demand (COD) and the removal of other substances such as heavy metals. Organic wastes were reported as excellent sources of energy-rich organic C-molecules as well as of macro- and micro-nutrients for microalgae cultivation. Microalgae can remove heavy metals from water and can utilize the waste for growth. Some literature can be referenced to further enhance the readability of this paper.
7. Please further improve the reference information, such as reference 28 Supplementation, D.M.; Functions, A.; Hens, A.L.
Author Response
We have responded to reviewer 3 comments as follows:
Reviewer comments 1: This paper collates an overview of existing algal applications and current market scenarios for microalgal products as foods and feeds along with of relevant legislative requirements concerning their use. This is a good work and well written. Below are some suggestions.
Response: We would like to thank the reviewer for their time and effort reviewing our paper.
- This in an interesting work and minor revision is recommended. The title is not specific enough. This paper is mainly about food and feed, and it is recommended to modify the title according to the main content.
Response: We thank the reviewer for their suggestion. We have revised the title as follows: “Applications of microalgae in foods, pharma and feeds and their use as fertilizers and biostimulants: Legislation and regulatory aspects for consideration”
Reviewer comment: Introduction. Moreover, they are known for their potential use as biofuels in renewable energy, as well as ingredients for use in biostimulants, cosmetic compounds, and healthcare products. Microalgae is an important raw material of biofuel, which has attracted more and more attention. Some recent literature can be referenced to improve the quality of this work, such as "Lipid accumulation by a novel microalga Parachlorella kessleri R-3 with wide pH tolerance for promising biodiesel production".
Response: We agree with the reviewer and we have now revised this text to read as follows: “Moreover, they are known for their potential use as biofuels in renewable energy [9], as well as ingredients for use in biostimulants [10], cosmetic compounds [11] and healthcare products. For example, Lipid accumulation by a novel microalga Parachlorella kessleri R-3 with wide pH tolerance is a promising strain for potential biodiesel production [12].”
Reviewer comment: Table 1. "sp." should be shown in italic form. Please check the whole manuscript.
Response: We have checked the document and the sp. is now in italics throughout the text.